# Minimal barriers to invasion during human colorectal tumor growth

Marc D. Ryser[1,2,3 ✉], Diego Mallo [4], Allison Hall[5], Timothy Hardman[6], Lorraine M. King[6], Sergei Tatishchev[7], Inmaculada C. Sorribes[2], Carlo C. Maley [4], Jeffrey R. Marks[3,6], E. Shelley Hwang[3,6] & Darryl Shibata[7 ✉]

Intra-tumoral heterogeneity (ITH) could represent clonal evolution where subclones with greater fitness confer more malignant phenotypes and invasion constitutes an evolutionary bottleneck. Alternatively, ITH could represent branching evolution with invasion of multiple subclones. The two models respectively predict a hierarchy of subclones arranged by phenotype, or multiple subclones with shared phenotypes. We delineate these modes of invasion by merging ancestral, topographic, and phenotypic information from 12 human colorectal tumors (11 carcinomas, 1 adenoma) obtained through saturation microdissection of 325 small tumor regions. The majority of subclones (29/46, 60%) share superficial and invasive phenotypes. Of 11 carcinomas, 9 show evidence of multiclonal invasion, and invasive and metastatic subclones arise early along the ancestral trees. Early multiclonal invasion in the majority of these tumors indicates the expansion of co-evolving subclones with similar malignant potential in absence of late bottlenecks and suggests that barriers to invasion are minimal during colorectal cancer growth.

[1] Department of Population Health Sciences, Duke University Medical Center, Durham, NC, USA. [2] Department of Mathematics, Duke University, Durham, NC, USA. [3] Duke Cancer Institute, Durham, NC, USA. [4] Arizona Cancer Evolution Center and School of Life Sciences, Arizona State University, Tempe, AZ, USA. [5] Department of Pathology, Duke University Medical Center, Durham, NC, USA. [6] Department of Surgery, Duke University Medical Center, Durham, NC, USA. [7] Department of Pathology, University of Southern California Keck School of Medicine, Los Angeles, CA, USA. ✉email: marc.ryser@duke.edu; dshibata@usc.edu

Direct observation of the clonal evolutionary process in human tumors is difficult. However, intratumor heterogeneity (ITH) is common in human tumors and the ancestral information recorded by subclonal mutations can be used to reconstruct their growth[1–3]. The interpretation of the ancestral information encoded by ITH is critical because knowledge about tumor initiation and progression informs effective strategies to prevent, detect, and treat human cancers[4–6].

Previous studies of ITH focused primarily on multiregional bulk sampling of tumors. While this approach enables mapping of ITH during lateral growth, it lacks the spatial and phenotypic resolution needed to characterize ITH during the critical downward growth (invasion) that confers tumor stage and prognosis. In this study, we show that phenotypic, ancestral, and topographic information can be merged after saturation microdissection and deep resequencing of tumor sections to characterize the phylogeography of invasion. By localizing subclones directly onto microscope sections we are able to infer the ancestry of each tumor region and the final histologic phenotypes of the subclones.

The path from start of superficial tumor growth to invasion of deep tissue is generally thought to follow a multistep progression[7]. In this model, increasingly fit subclones expand in the superficial layers until an evolutionary bottleneck event gives rise to a subclone that penetrates the muscularis mucosae (MM) and invades the deeper tissue (Fig. 1a, left). Consequently, the subclone topography on microscope slides is expected to consist of contiguous subclones layered horizontally by phenotype (Fig. 1b, left), with a single late subclone acquiring the ability to invade the deep tissue (Fig. 1c, left).

Here, we show that the mutation topographies on microscope slides in 12 human colorectal tumors do not support an evolutionary bottleneck model of invasion. To the contrary, we find that the majority of tumors show evidence of multiclonal invasion (Fig. 1a, right), where multiple jigsaw arrayed subclones span from superficial to invasive regions (Fig. 1b, right). Through reconstruction of ancestral trees we find that invading subclones arise early during growth, suggesting growth dynamics that are expected under branching evolution[1,8] (Fig. 1c, right) as opposed to a multi-step progression with late invasion (Fig. 1c, left). In particular, the observed phylogeographies are consistent with the near-neutral Big Bang model of tumorigenesis, a type of branching evolution that has been extensively documented in these tumors[9–11].

In summary, and similarly to a recent study in early stage breast cancers[12], our results indicate that multiclonal invasion is common in colorectal tumors and indicate that there are only minimal barriers to invasion after the start of tumor growth.

## Results

**Saturation microdissection of tumor slides**. A prerequisite for mapping the topography of tumor subclones is the ability to physically isolate regions small enough to contain a single subclone, that is a group of cells sharing the same genotype. Fortunately, prior studies indicate that single colorectal tumor glands are defined subclones because they are clonal for the public (found in all sampled regions) and private (found in a strict subset of sampled regions) mutations identified through exome sequencing of bulk samples[9]. To leverage this gland level clonality, we microdissected individual spots (small regions consisting of 2–5 adjacent glands each) from multiple regions of microscope slides (Fig. 2a) obtained from 12 human colorectal tumors (Table 1, Supplementary Table 1). From each primary tumor, two (one for tumor R) arbitrarily located sections were collected and a median of 13 spots (range: 6–20) per slide were microdissected.

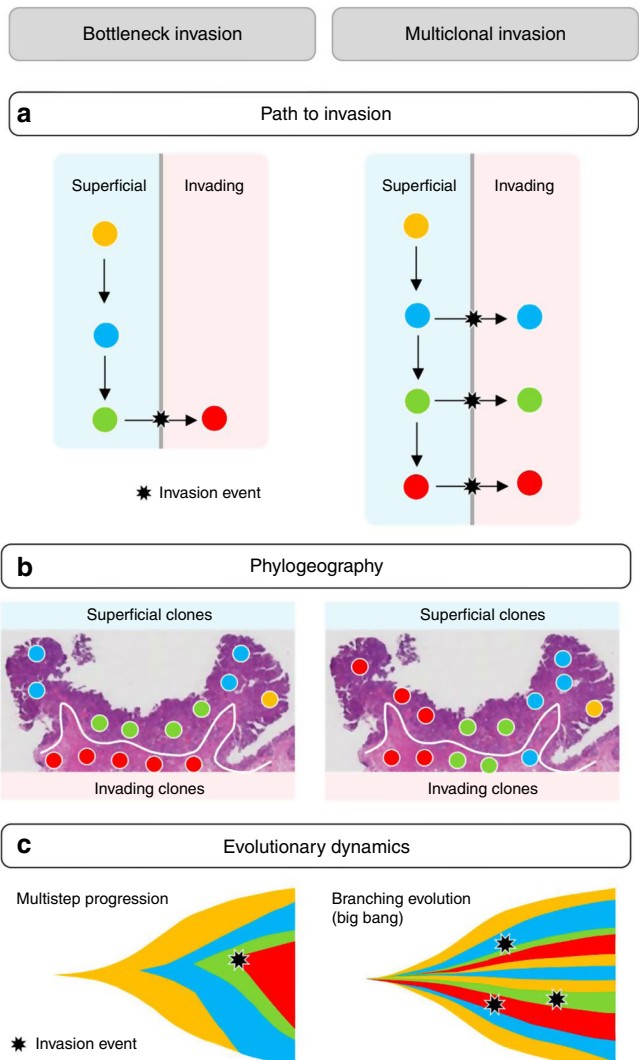

**Fig. 1 Bottleneck vs Multiclonal Invasion.** Topographic and phenotypic subclone distributions generally differ between the bottleneck (left) and multiclonal invasion (right) models. **a** In the bottleneck model (left), the tumor evolves in superficial regions and invasion constitutes an evolutionary bottleneck where only the more advanced subclones eventually invade the stroma. In the multiclonal invasion scenario (right), co-existing subclones have a similar malignant potential to invade the stroma. **b** Left: if growth occurs through bottlenecks, subclones should be arranged horizontally with respect to phenotype, with a smaller number of more advanced subclones (red) with invasive phenotypes. Right: under multiclonal invasion, subclones have similar malignant potential; expansile growth occurs radially in all directions resulting in vertical subclones columns with both superficial and invasive phenotypes in the final tumor. **c** Left: the bottleneck model is compatible with the classic multi-step progression of cancer evolution whereby the invasive clone arises late (star). Right: the multiclonal invasion scenario is compatible with branching evolutionary models with multiple co-evolving subclones, such as predicted by the Big Bang model.

The average nearest neighbor distance between spots was 2.9 mm, with an average pairwise distance of 8.4 mm between dots on the same slide. Additional sections from metastatic sites were obtained and microdissected for three tumors (J, C, H).

**Distinct subclones are present in small tumor regions.** For each tumor, a panel of candidate mutations (median number of loci:

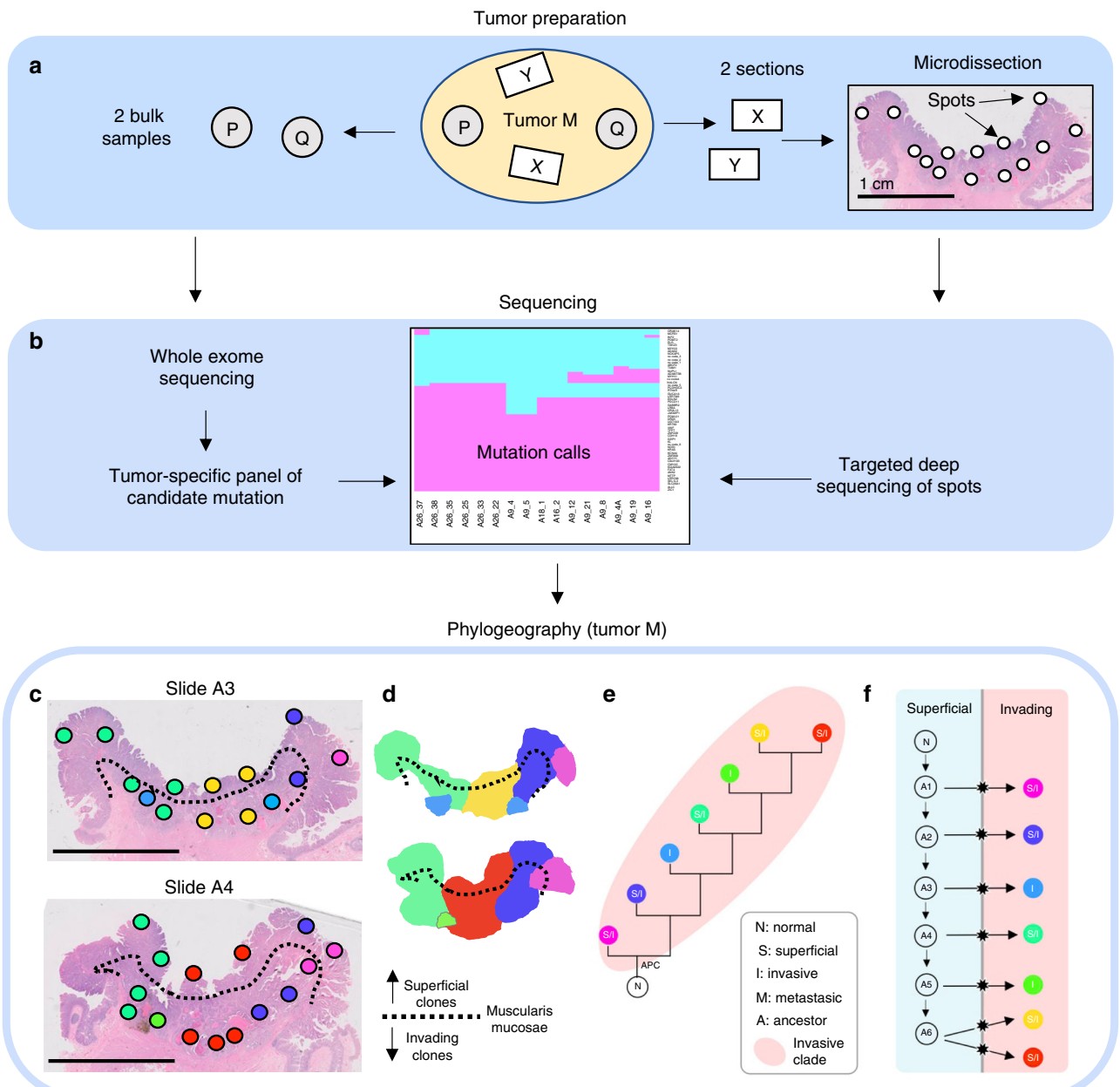

**Fig. 2 Study design. a** From each tumor, two bulk samples from opposite sides were obtained (P, Q) and two microscope sections (X, Y) were collected during routine clinical work (arbitrary locations). From the microscopic sections, small spots (2–5 glands each) were microdissected using SURF technology. **b** After whole-exome sequencing of the bulk samples P and Q, tumor-specific mutation panels with ~50 candidate mutations were designed. For each spot from the sections X and Y, targeted deep re-sequencing of the candidate mutations was performed. Because individual spots were clonal, mutations were called using a noise threshold for variant allele frequencies (VAF) at 5%. **c** Spot genotypes (here for tumor M) were identified and localized on the sections and sections were annotated for superficial and invasive regions (scale bar: 1 cm). **d** Invasion was defined as migration below the muscularis mucosae (dotted line). Contiguous clone maps were derived from the spot topographies. **e** Maximum parsimony algorithms were used to reconstruct phylogenetic trees for the subclones. **f** The ancestry of invasive subclones was reconstructed under the assumption that cells can migrate from superficial to invasive regions, but not vice versa.

47, range: 13–110) was derived from a previous whole-exome sequencing study[9] (Fig. 2b). Deep (mean depth ~9000x; minimum depth: 20x) targeted re-sequencing of the tumor-specific mutation panels was performed. Variant allele frequencies (VAF) of private mutations were comparable to those of public mutations (Supplementary Fig. 1, Supplementary Data 1), consistent with the presence of a single subclone in each spot. Indeed, of 325 analyzed spots, only four harbored multiple subclones as indicated by low private mutation VAFs and homoplasy

(Supplementary Table 1). The remaining 321 clonal spots were assigned a genotype, enabling the mapping of subclone topographies (Figs. 2c, 3a, e, 4a, f, Supplementary Fig. 2a) and reconstruction of maximum parsimony phylogenies (Figs. 2e, 3c, g, 4d, i, and Supplementary Fig. 2c). The jigsaw-arranged subclones varied in their topographic spread, with a median of 3.5 spots per subclone (range: 1–27; Fig. 5a). There was a median of three subclones per slide (range: 1–7; Fig. 5b) and a median of five subclones per tumor (range: 1–9; Fig. 5c).

**Table 1 Data summary.**

| ID | Spots | Loci | Subclones | Phenotypes[a] S/I/S-I | Monophyletic Inv | Early Inv | Multiclonal Inv | Inv events | Mixing[b] |
|---|---|---|---|---|---|---|---|---|---|
| C | 16 | 57 | 9 | 8/0/0 | Yes | Yes | No | 1 | No |
| D | 33 | 57 | 5 | 3/0/2 | No | Yes | Yes | 2 | Yes |
| E | 27 | 110 | 6 | 2/2/2 | No | Yes | Yes | 3 | Yes |
| F | 31 | 53 | 5 | 1/1/3 | No | Yes | Yes | 4 | Yes |
| H | 39 | 30 | 3 | 0/1/2 | Yes | Yes | Yes | 3 | No |
| J | 27 | 28 | 1 | 0/0/1 | Yes | Yes | No | 1 | No |
| K | 29 | 41 | 9 | 10/0/0 | NA | NA | NA | NA | No |
| M | 29 | 13 | 7 | 0/2/5 | Yes | Yes | Yes | 7 | Yes |
| R | 15 | 58 | 3 | 0/1/2 | Yes | Yes | Yes | 3 | Yes |
| T | 26 | 35 | 5 | 0/1/4 | Yes | Yes | Yes | 5 | Yes |
| U | 23 | 29 | 4 | 1/1/2 | No | Yes | Yes | 3 | Yes |
| W | 30 | 68 | 7 | 1/0/6 | No | Yes | Yes | 6 | Yes |
|   | Total 325 | Total 579 | Median 5 | Total 26/9/29 | 6/11 | 11/11 | 9/11 | Median 3[c] | 8/12 |

*CRC carcinoma, Adx adenoma, S superficial, I invasive, S/I mixed S and I, Inv invasion, NA not applicable.*
[a]Primary tumor only.
[b]Early mixing as inferred in ref. [9].
[c]Multiclonal tumors only.

**Most subclones share superficial and invasive phenotypes**. The merged ancestral, topographic, and phenotypic information enabled us to test whether observed ITH was more consistent with late bottleneck or multiclonal invasion (Fig. 1). To this end, we first combined histopathologic examination and spatial registration to classify each microdissected spot as either superficial, invasive, or metastatic (see the "Methods" section for details). We found that subclones with multiple spots were commonly arrayed in contiguous vertical columns spanning from superficial to invasive regions (Figs. 2d, 3b, f, 4b, g, Supplementary Fig. 2b). Across 10 carcinomas with at least one invasive spot in the primary tumor (thus excluding tumor C), the majority of subclones (29/46, or 63%) had a mixed phenotype, that is they covered both superficial and invasive regions (Fig. 5d). Further restricting the analysis to the 37 subclones with more than one spot, the fraction of mixed subclones increased to 76%. Subclone phenotypic heterogeneity is visualized on t-SNE plots where genotypes but not phenotypes cluster together (Fig. 5e, Supplementary Fig. 3). In summary, instead of subclones layered horizontally by phenotype (Fig. 1b, left) as predicted by multistep progression, the majority of subclones were of mixed phenotype (Fig. 1b, right).

**Fingerprints of multiclonal invasion**. Next, we sought to reconstruct the invasion dynamics in the 11 invasive carcinomas. Among six of these, invading subclones (of either invasive or mixed phenotype) formed a polyphyletic group in the ancestral tree, indicating multiclonal invasion (Table 1; Figs. 2e, 3c, g, 4d, i, and Supplementary Fig. 2c). Further assuming that once invasive, subclones did not migrate back across the MM into superficial regions, we were able to reconstruct the number of invasion events for each tumor (Figs. 2f, 3d, h, 4e, j, and Supplementary Fig. 2d). Of the 10 multiclonal CRCs (tumor J was monoclonal), only tumor C had a single invasion event compatible with an evolutionary bottleneck. The remaining nine carcinomas had undergone multiclonal invasion, with a median of three invasion events per tumor (range: 1–7; Table 1) and exhibited comparable levels of genotypic diversity in superficial and invasive regions (Fig. 5f). In summary, evidence of multiclonal invasion in 9 of 11 carcinomas suggests that the observed ITH is consistent with multiple invading subclones during growth (Fig. 1c, right).

**Similar dynamics for early and late ancestral subclones**. We examined whether late branching subclones may experience stepwise increases in fitness and thus more advanced phenotypes. To this end, we subdivided subclones into early or late categories based on whether they arose during the first or second half of tumorigenesis, as measured by the distance from the root in the maximum parsimony trees (Supplementary Fig. 4c). Among carcinomas, 21 and 34 subclones were found on early and late branches, respectively. Early invasion events were found in all 11 CRCs (Table 1), and there was no size difference between early and late subclones (Wilcoxon rank-sum test: $p = 0.7$), which both had a median size of 4 spots per clone (Fig. 5g). We note that we only analyzed a limited number of spots in each tumor, and it is likely that the resulting phylogenies do not capture the complete ancestries of the tumors. However, because the ordering of subclones on the sampled tree is the same as on the complete ancestral tree (i.e., from the zygote to the tumor), we can nevertheless conclude that there was no evidence of later subclones being more phenotypically advanced than earlier subclones.

**Jigsaw subclone topography**. Because tumors grow by gland splitting or fission[13], neighboring glands should be related and subclones expand continuously. Consistent with these predictions, most subclones with more than 2 spots in a slide were spatially contiguous (41 of 46, or 89%). Although contiguous subclones are expected with both stepwise progression and branching evolution, the phylogenetic relatedness of neighboring subclones can provide further delineation between the models. First, we found that the normalized genetic distance between adjacent subclones had an empirical distribution (Fig. 5h) that was not significantly distinct from a uniform distribution ($p > 0.1$, two-sided Kolmogorov-Smirnoff test). Thus, and in contrary to what would be expected under stepwise progression, physically adjacent subclones were essentially randomly sampled from the ancestral tree. Second, the relationship between spatial and genetic distances of spots within slides was heterogeneous, with Pearson correlations ranging from −0.2 to −0.8 (median: 0.37) across the 12 tumors (Supplementary Fig. 6). Third, we found that the pairwise distances of subclones in the same slide were similar to those of subclones from two distinct slides in the same tumor. With the exception of tumor C, there were no differences between within- and between-slide pairwise distances (Supplementary Fig. 7). Taken together, these observations suggest that the co-evolution of equivalent subclones may have been preceded by early subclone mixing[9], leading to the observed jigsaw arrangements of subclones in the final tumor as evidenced by a

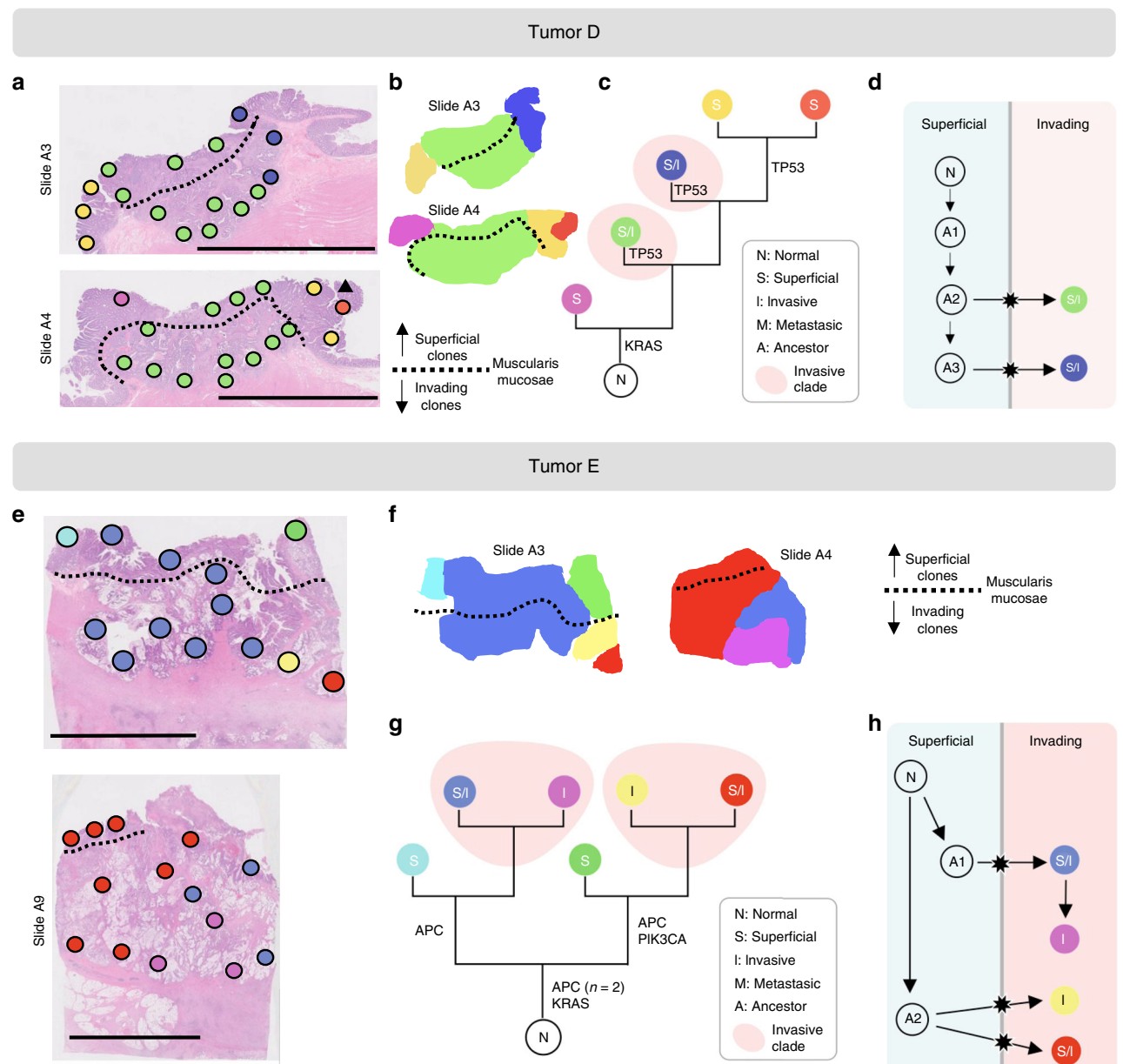

**Fig. 3 Phylogeographies of tumors D and E. a**, **e** Spot genotypes were localized on the sections and sections annotated for superficial and invasive regions (scale bar: 1 cm). Invasion is defined as migration below the muscularis mucosae (dotted line). **b**, **f** Contiguous clone maps reveal vertically arranged subclones that share superficial and invasive phenotypes. **c**, **g** Maximum parsimony algorithms were used to reconstruct phylogenetic trees for the subclones. **d**, **h** The ancestry of invasive subclones were reconstructed under the assumption that cells migrate from superficial to invasive regions, but not vice versa.

lack of consistent correlations between spatial and ancestral relatedness.

**Recognizing stepwise progression.** Although a lack of clear morphologic boundaries or size differences between the jigsaw arrayed subclones is consistent with tumorigenesis by near-neutral subclones, the presence of clonal selection is difficult to rule out[14,15]. Indeed, a key challenge in recognizing selection during tumor growth is that the relative fitness of a subclone needs to be very high in order to outrun its neighboring, neutrally expanding subclones[9,16,17]. Focusing instead on established canonical driver mutations, we found that 11 of 12 tumors had at least one public driver mutation (tumor H had none). Two tumors (D and E) had additional private driver

mutations that may have conferred an increase in fitness to individual subclones during expansion. In tumor D (Fig. 3a–d), the largest subclone had its own private *TP53* mutation (C238Y) and had the histologic features of stepwise progression with central invasion and relatively distinct morphologic boundaries. However, this subclone did not arise in stepwise fashion from the surrounding superficial subclones as indicated by the presence of two superficial subclones that arose later in the ancestral tree and harbored their own private *TP53* mutations (V143A, R248Q). In tumor E (Fig. 3e–h), both early branches had their own private driver mutations (*APC* and *PIK3CA*, and *APC*). In summary, in the analyzed tumors we found only limited evidence of selection through the acquisition of private driver mutations.

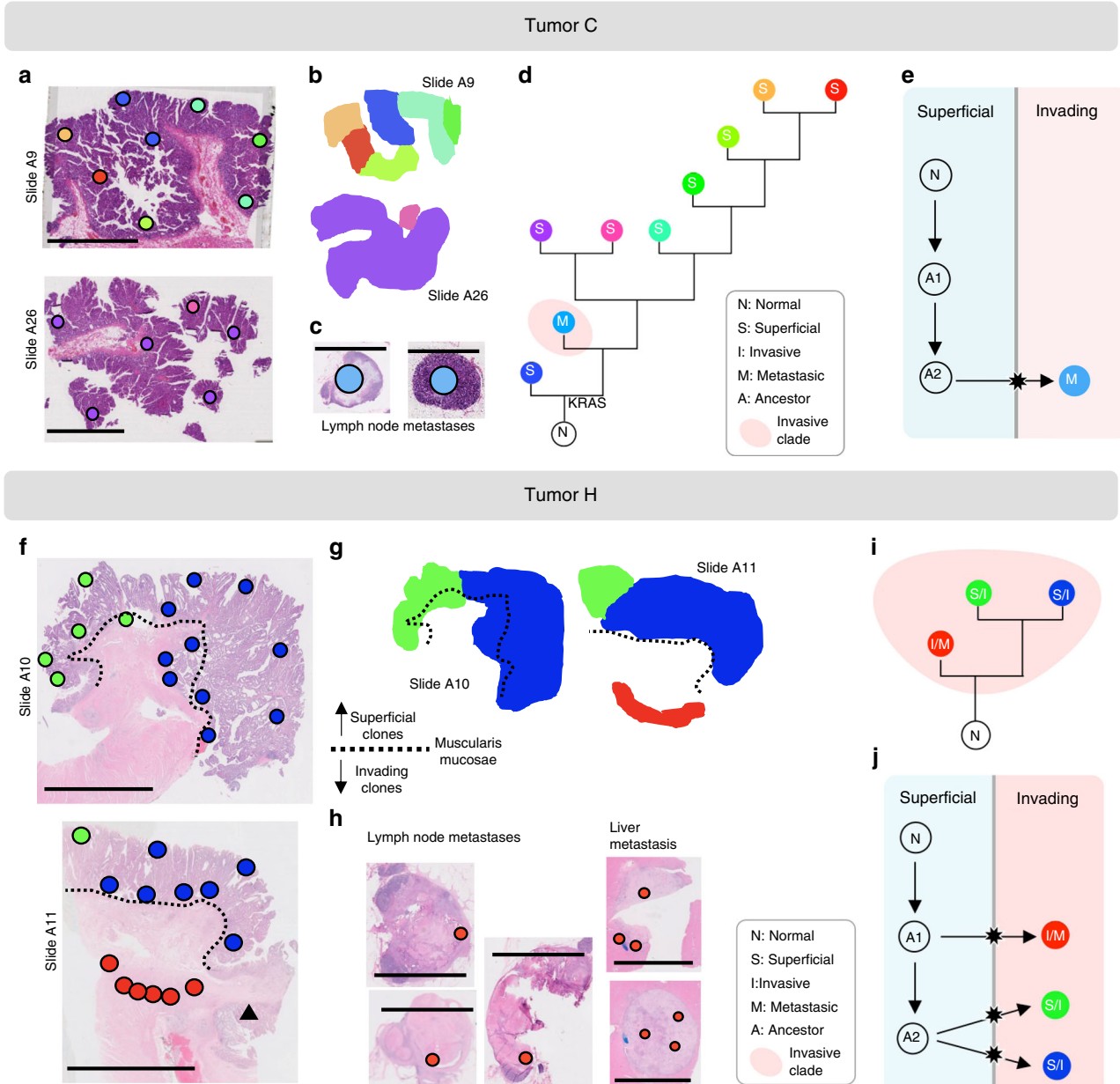

**Fig. 4 Microscopic metastases and early branching in tumors C and H. a, f** Spot genotypes were localized on the sections and sections annotated for superficial and invasive regions (scale bar: 1cm). In tumor, H, a deeply invasive ribbon of cells (red, slide A11) shared the genotype of the metastases, yet did not share the private mutations of the overlying superficial portions of the primary. **b, g** Contiguous clone maps were derived from the spot topographies. **c, h** Additional spots in the lymph nodes and liver metastases were microdissected (scale bar: 0.5 cm). **d, i** Maximum parsimony algorithms were used to reconstruct phylogenetic trees for the subclones. In tumor C, the two lymph node spots shared genotypes and branched early; the metastatic genotype was not found in the primary tumor. **e, j** The ancestry of invasive subclones was reconstructed under the assumption that cells migrate from superficial to invasive regions, but not vice versa.

**Invasion and metastasis can occur very early**. Finally, we looked for direct evidence of stepwise progression by examining tumors with microscopic presence of small metastatic foci which upgrade clinical staging, confer poorer prognosis, and represent the zenith of stepwise progression. We analyzed three CRCs (J, C, H) with small microscopic metastases in the lymph nodes and liver (H only) (Fig. 4, Supplementary Fig. 2). In all cases the metastatic spots were genotypically identical within a patient and lacked any of the private mutations of the primary, indicating early divergence. In two cases (J, C), the microscopic metastatic foci had all the public mutations; in tumor H, the microscopic metastases diverged even earlier as evidenced by the lack of 10 of 22 public

mutations present in the primary tumor. Of note, none of the canonical CRC drivers were missing in the metastatic subclones. In summary, the acquisition of more advanced phenotypes in these tumors occurred along the earliest ancestral branches, indicating that invasion and metastasis may occur very early during growth.

## Discussion

Tumors are heterogenous populations of cells with different phenotypes and multiregional sampling has revealed that genetic ITH is also very common. However, there is uncertainty about

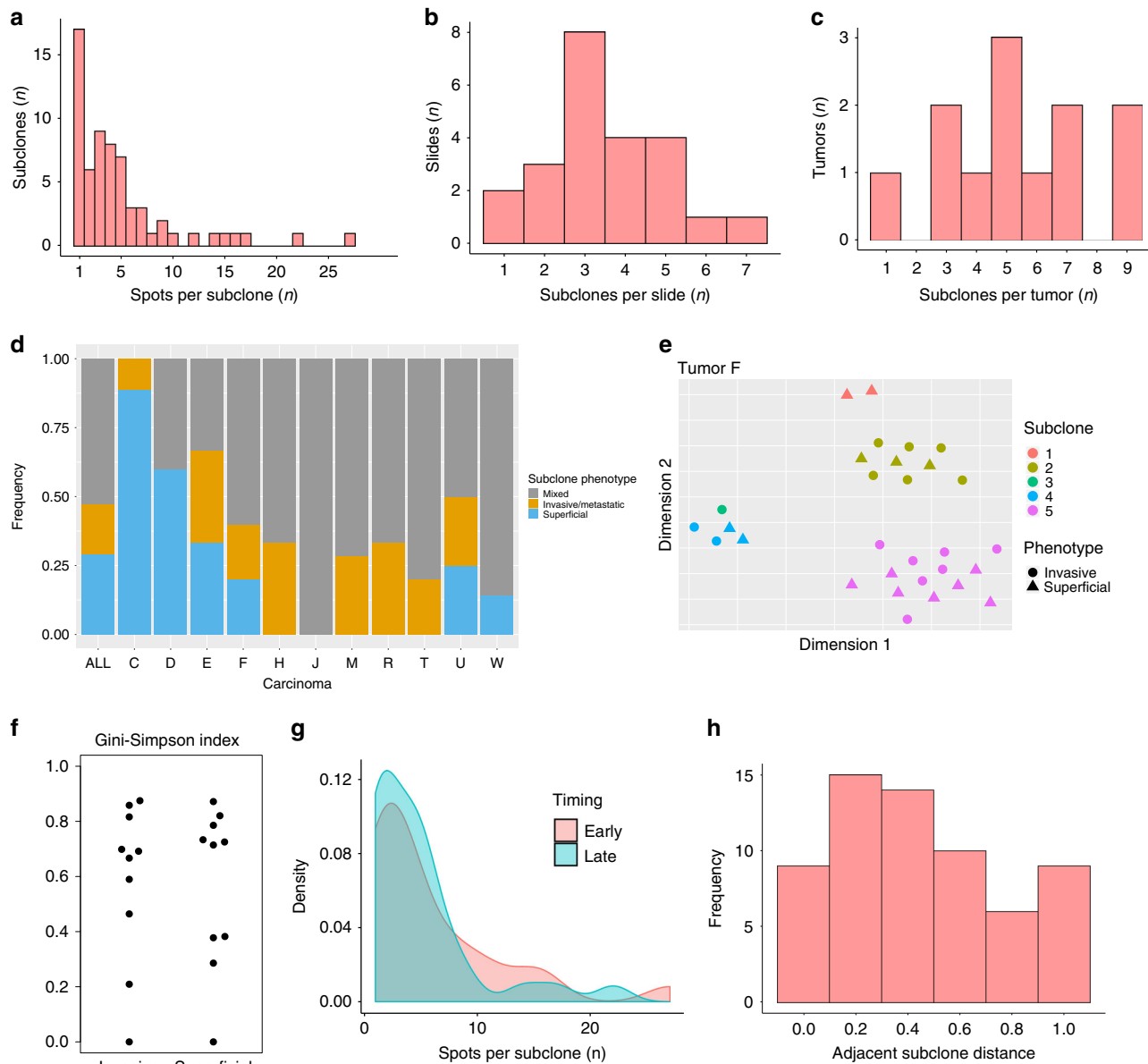

**Fig. 5 Summary statistics. a** Across all 12 tumors, the number of spots per subclone ranged from 1 to 27 (median: 3.5). **b** Across all 23 microscopic tissue slides, the number of subclones per slide ranged from 1 to 7 (median: 3). **c** The number of subclones per tumor ranged from 1 to 9 (median: 5). **d** With the exception of tumor C, all tumors had subclones of mixed phenotype, i.e., the same genotype was found in superficial and invasive/metastatic spots of the tumor. **e** Nonlinear embedding (t-SNE) of the high-dimensional sequencing data for tumor F illustrates that spots cluster by genotype (colors) but not by phenotype (shapes). t-SNE plots for all tumors in Supplementary Fig. 3. **f** For each tumor (except C and K), subclone diversity in superficial and invasive regions was quantified using the Gini-Simpson index. There was no significant difference between the two groups ($p = 0.88$, two-sided paired t-test). **g** Early and late subclones (as measured by relative distance from the root of the ancestral tree, see Supplementary Fig. 4) had similar subclone distributions. **h** The normalized phylogenetic distance (number of separating internal nodes) between physically adjacent subclones was broadly distributed.

the number and size of subclones, how they are arranged spatially, and whether subclones with different genotypes have distinct phenotypes. Here we addressed these questions through saturation microdissection and targeted resequencing of tumor sections on microscope slides, assessing the spatial distributions of subclones and their histologic phenotypes.

Consistent with the classical model of stepwise progression, colon cancers have a hierarchy of progressing phenotypes, from superficial to deeply invasive cells. This downwards spatial histologic progression also corresponds to clinical staging criteria, where more advanced cancers of poorer prognosis are defined by more deeply invasive or metastatic tumor cells. Similar to the phenotypic

hierarchy, genetic ITH can be cast as a hierarchy of early and late branching subclones along the phylogenetic tree. Under stepwise progression, the phylogenetic hierarchy is expected to directly reflect histologic barriers to progression (bottlenecks). More precisely, early subclones with limited malignant capabilities can spread horizontally, but not downward into the deep tissue, whereas later subclones have more malignant capabilities that enable deeper invasion and metastasis. Hence if the ancestral trees represent stepwise selection, more invasive subclones should arise late on an ancestral tree (Fig. 1c, left). Saturation microdissection can directly test whether early and late subclones co-localize with superficial and invasive phenotypes, respectively.

Contrary to the expectations from the stepwise progression model, we found that most tumor sections contained multiple, vertically arranged and millimeter-sized subclones that shared invasive and superficial phenotypes. In 9 of 11 carcinomas, invasive phenotypes arose multiple times (multiclonal invasion) from both early and late branches, which suggests branching evolution rather than stepwise progression with invasive bottlenecks. Only 2 of 11 carcinomas harbored private driver mutations that indicate possible selection during growth. One of them (tumor D) had many features of stepwise progression but its dominant subclone arose early in its ancestral tree and not directly from neighboring superficial subclones. Therefore, while it is possible that more direct progenitor lesions were missed during sampling or may have been replaced during expansion of the dominant clone, the data do not support a late bottleneck to invasion in this tumor.

The observed dissociation of phenotypic and genotypic progression during tumor growth is consistent with branching evolution or Big Bang tumorigenesis[11] where the founder cell already starts at a high fitness level and has the driver mutations sufficient for rapid growth (Fig. 1b). Aligned with this scenario, most driver mutations in this and other studies[9,18] were clonal, that is present in all sampled areas. The subclones were defined largely by passenger mutations that show little evidence of selection in cancers[19–21], and unsurprisingly, the subclones mapped by saturation microdissections were present in both the invasive and non-invasive components without distinct boundaries between those components.

Genetic subclones with multiple phenotypes imply plasticity, where the same cell can adopt multiple phenotypes depending on its microenvironment[16,22]. Phenotypic plasticity allows for rapid growth because the founder cell and its progeny can readily adapt and have either invasive or superficial phenotypes depending on their locations within the tumor. Multiregional sampling of the same tumors showed preferentially conserved epigenomes as evidenced by preservation of unmethylated promoters and enhancers between opposite tumor sides[10]. Interestingly, phenotypic plasticity is already present in normal colon where differentiated cells share similar epigenomes and cell phenotype is determined by the microenvironment[23].

Given a founder cell with the driver mutations and an epigenome sufficient for rapid growth and adaptation, invasion can occur very early and by multiple, essentially identical subclones. The ancestral trees inferred in this study likely reflect very early events in the first tumor gland because subclonal mutations that occur later would not be detectable by exome sequencing[9,15]. Consequently, the final locations of subclones within a tumor depend primarily on cell movement or mixing during early growth[9], which in turn leads to the observed jigsaw-arranged subclone patterns that exhibit little correlation between physical and ancestral distances.

The vertically arranged subclone patterns can help explain why invasive and metastatic phenotypes arise early on ancestral trees. Indeed, with phenotypic plasticity, the initial direction of cell growth determines subclone phenotype because downwards (deep) growing subclones have greater physical access to stroma and vasculature than upwards (superficial) growing subclones. This in turn leads to star-shaped ancestral trees[24] where invasion and metastasis can begin from the start of growth rather than being constrained by a bottleneck until superficial tumor growth is completed. The early ancestral branching of metastases is a common observation in human cancers[18,25–28], and as corroborated by our findings, likely occurs very early when the primary tumor consists of less than one million cells[29].

In summary, by merging ancestral, topographic, and phenotypic information from microdissected slides, we developed an experimental approach that provides direct insights into the growth of individual human tumors. Although the studied tumors are consistent with single Big Bang expansions governed by branching evolution, there is considerable heterogeneity within and between tumors, which may reflect complex microenvironmental interactions or selection at scales smaller than the current millimeter resolution[5]. By increasing the number of micro-dissected spots and targeting private mutations with lower allelic frequencies, characterization of heterogeneity could be further refined and used to reconstruct how lethal human cancers start to grow. Indeed, mass screening programs have led to widespread overdiagnosis and overtreatment of small tumors[30], and tackling these issues requires effective discrimination between indolent and lethal lesions at time of diagnosis. As such, the reconstruction of patient-specific ITH patterns from routine diagnostic materials as illustrated in this study may further enhance the delivery of effective personalized oncology.

## Methods

**Tumor samples**. The 12 colorectal tumors (Table 1) were previously studied[9] and collected as excess tissues in the course of routine clinical care (informed consent exemption). The studies were approved by the Institutional Review Board at the University of Southern California Keck School of Medicine.

**Saturation microdissection and deep resequencing**. Microdissection was performed using selective ultraviolet light fraction (SURF)[31]. Briefly, microscopic sections were placed on plastic slides, lightly stained, and small ink dots were placed directly over 2–5 tumor glands using a micromanipulator (Supplementary Fig. 5). Unprotected DNA was destroyed by 3–4 h of short-wave ultraviolet light irradiation. The spots on the plastic slide were cut out and individually placed in a microfuge tube for DNA extraction (TE and Proteinase K, 60 C for 4 h, then 98 °C for 10 min), using a pipette tip to remove the ink dot from the plastic slide. AMPure XP beads (Beckman Coulter) were added (1.2×) to extract the DNA. PCR (35–40 cycles) was directly performed on the dried beads, using custom AmpliSeq primers for tumor-specific single nucleotide variants (SNVs). The latter comprised both public (present in all samples) and private (present in strict subset of samples only) mutations as identified in a previous study[9]. Barcoded libraries were made (One-step, Qiagen) and run on MiSeq or NextSeq Illumina sequencers. Average coverage was ~9000x with a minimum of 20 reads per spot required for the genotyping. The initial mutation calling threshold was a VAF of 0.05. The complete data set (VAFs) is found in the Supplementary Data file.

**Phenotypic classification**. Phenotypic classification of micro-dissected spots was performed by two experienced board-certified pathologists (DS and AH). After independent classification by the two pathologist, discordant labels were resolved through consultation with a third board-certified pathologist who specializes in GI pathology (ST). Histopathologic examination of the microscope slides was used to classify each spot as either superficial, invasive, or metastatic. All spots obtained from lymph nodes or distant organs were labeled as metastatic. To distinguish between superficial and invasive spots on slides from the primary tumors, the following rules were applied: (i) in slide sectors where the MM was intact (local Tis stage), spots above the MM were labeled as superficial; and (ii) in slide sectors where the MM was no longer present (local T1+ stage), spots were labelled as superficial if in close proximity to the surface (lumen), and as invasive otherwise. The trajectory of the MM was indicated where still intact. In sectors with degraded MM, the most likely path of the original MM was indicated; this was done for visualization purposes only and had no bearing on spot classification.

**Computer code**. The analyses described below were all performed with the software R (version 3.5.3, R Foundation for Statistical Computing, Vienna, Austria).

**Phylogenetic tree construction**. Prior to ancestral reconstruction, two additional pre-processing steps were implemented for each tumor: (i) if a private mutation was found in only one spot and with a VAF<6%, it was interpreted as noise and classified as absent; (ii) if a mutation was present in more than 90% of all spots in the tumor, missingness of the mutation in the remaining spots was deemed a false negative finding and the mutation was classified as public. After pre-processing, each spot was assigned a genotype, that is a binary vector indicating the presence and absence of mutations. Unique genotypes were labelled as subclones and consequently, spots that share the same genotype belong to the same subclone. To derive the phylogenetic relationship between subclones, maximum parsimony trees were constructed using Nixon's ratchet algorithm[32] as implemented in the *phangorn* package in R. Bootstrap sampling (*n* = 1000) was performed to quantify tree building confidence. Trees were constructed under the principle of homoplasy avoidance by assuming that each mutation is acquired only once, and that SNVs

are not selectively reversed during evolution (both events have a very low probability). Homoplasy avoidance was achieved as follows: (i) by removal of spots whose VAF spectrum indicated admixture of two or more subclones (n = 4); (ii) reclassification of false-positive and false negative calls (n = 12). See Supplementary Table 1 for details on homoplasy avoidance.

**t-SNE plots**. To visualize genotypic and phenotypic clustering properties of individual spots (Fig. 5e, Supplementary Fig. 3), we used t-distributed stochastic neighbor embedding, or t-SNE[33], as implemented in the package *Rtsne* in R.

**Normalized distance between adjacent subclones**. This measure was computed to evaluate the genetic distance between spatially adjacent subclones (Fig. 5h). For each tumor (except J, which only had one subclone), adjacent subclones were identified based on the mutation topographies. For each pair of adjacent subclones, the number of separating internal nodes was counted. A normalized metric was obtained by subtracting 1 and dividing the resulting integer by the maximum distance between any two tips on the tree.

**Subclone diversity**. Diversity of subclones in superficial and invasive regions of the tumors (Fig. 5f) were quantified using the Gini-Simpson index (GS), which is defined as the probability that in a random sample of two spots (without replacement), the corresponding subclones are different, or

$$\text{GS} = 1 - \frac{1}{N(N-1)} \sum_{i=1}^{k} n_i(n_i - 1), \tag{1}$$

where $N$ is the number of spots in the region of interest, $k$ is the number of subclones, and $n_i$ is the number of spots in subclone i.

**Spatial vs. genetic distances**. The pairwise spatial distances between spots on the same microscope sections were manually recorded. The genetic distance between spots was obtained, after processing (see above), as the $L_1$-distance (Manhattan distance) between dichotomized VAF vectors. To quantify the correlations between genetic and spatial distances (Supplementary Fig. 6), the Pearson correlation coefficients between the two measures were computed for each tumor.

**Reporting summary**. Further information on research design is available in the Nature Research Reporting Summary linked to this article.

## Data availability
The whole-exome sequencing data have been deposited in the Short Read Archive database under the accession code PRJNA602679. All the other data supporting the findings of this study are available within the paper and its supplementary information files or available from the corresponding author upon reasonable request.

## Code availability
The computer code used in this manuscript is available at https://github.com/mdryser/D5_Colon (MIT License).

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

## Acknowledgements
This work was supported in parts by grants from the Triangle Center for Evolutionary Medicine, National Institutes of Health (R00 CA207872, P30 CA014236, U54 CA217376, U2C CA233254, P01 CA91955, R01 CA170595, R01 CA185138, R01 CA140657), the National Science Foundation (DMS 1614838), CDMRP Breast Cancer Research Program Award (BC132057) and Arizona Biomedical Research Commission (ADHS18-198847).

## Author contributions
D.S. and M.D.R. designed the experiments, analyzed the data and wrote the manuscript. D.M., A.H., T.H., L.M.K., I.C.S., T.S., C.C.M., J.R.M., and E.S.H. assisted with data analysis, interpretation, and manuscript editing.

## Competing interests
The authors declare no competing interests.
