## [Peer Review File · Nature Communications]

Reviewers' comments:

Reviewer #1 (Remarks to the Author): Expertise in colorectal cancer

Ryser and colleagues describe an interesting multi-region sequencing study to analyze the invasion pattern of colorectal cancers. Multiple subclones are identified from mutations in a panel of targeted genes and from the spatial organization of the subclones the authors conclude that multiple subclones invade in parallel and the growth dynamics is neutral. I am, however, not sure if their data fully supports this conclusion.

To conclude that multiclonal invasion and neutral growth is the dominant mode of growth I believe that in a tumor the following spatial distribution of sub-clones should be observed: 1) Multiple clones should be observed both above and below the muscularis mucosae (the boundary for invasion used by the authors). 2) The genetic diversity should be similar above and below the muscularis mucosae.

Only in tumor M both of the above conditions are met, in all other tumors presented in the main manuscript either there is just one subclone present both above and below the muscularis mucosae or the genetic diversity is much lower below the muscularis mucosae (e.g. in tumor H the green clone might be marginally invasive but the blue clone clearly dominates the invasive area, indicating a selective benefit.)

In Fig. 1 the authors do introduce two models of growth and the expected spatial patterns of subclones. It would help if they quantify the expectations of these models, and quantify their observed data, to compare and assess which tumor fits in which model. Contrary to what the authors suggest in the discussion, I believe their work could greatly benefit from explicit modelling efforts. In particular, it would be very interesting to see the predictions from spatially resolved modelling in conjunction with their spatial genetic data, similar to e.g. Sun et al., Nature Genetics 2017 and Van der Heijden et al., PNAS 2019.

In addition it would help if the authors mention their definition of a subclone in the main manuscript.

Are the authors planning to make their data publicly available?

Reviewer #2 (Remarks to the Author): Expertise in clonal evolution

The manuscript entitled "Minimal barriers to invasion during colorectal tumor growth" by Ryser et al. presents targeted sequencing data from small colorectal cancer samples that the authors classify as superficial or invading. By placing the tumor areas on a phylogenetic tree and observing that invading areas are polyphyletic groups, the authors argue that invasion is not the trait of a unique subclone, but that many if not all cells are capable of it.

In my opinion, the topic addressed here is of great interest to the scientific community. Much remains to be learned about the evolution of tumors, and it is important and meaningful to investigate whether the invasive phenotype may be correlated with a distinct genotype. The first step is to determine whether invading cells belong to the same clone, which the authors attempt to do here. However, while the topic is very interesting, I am not convinced by the implementation of this study. My main concern is the lack of detail in the description of the histopathological work, which is the foundation for all conclusions drawn here.

Main concern:

Pathological review and categorization of samples: the methods section contains 3 lines of text on

the topic of "tumor samples". That is not sufficient for a paper whose main conclusion is based on pathological categorization of tumor areas into invading and superficial. Vastly more detail would be needed here. How were the tumor areas classified? Presumably by a GI pathologist? Which rules did this pathologist employ? The text mentions that the muscularis mucosae was used as a dividing line, but what about tumors in which the muscularis mucosae no longer exists because the entire area between the lumen of the colon and the stroma is occupied by tumor? The dashed dividing lines in many of the histological images are concerning in my opinion. For example, in figure 2i (both slide A3 and A4), the dividing line appears to be running straight through the epithelium and sometimes it even runs within the lumen... Many clones that are categorized as "invasive" by such a casually drawn line are superficial clones in my opinion (e.g. several of the green clones in slide A3/Figure 2). Similar concerns apply to other slides, e.g. slide A3 in Figure 2B. In the absence of a very detailed description of the analysis performed by the GI pathologist, I am doubtful that many of these clones are truly invasive, and not just contiguously growing tumor on the mucosal surface. Since the conclusions of the study rest so heavily on the superficial/invading categorization, the authors need to justify their categories in much more detail.

Comparatively minor concerns:

Given that all clones come from a very small sampled tumor area (corresponding to one or two slides), it is not clear to me how meaningful the "early/late" classification is. Presumably, given the constrained sampling area, these could all be "late" subclones that have recently evolved in this part of the tumor. This should at least be discussed in my opinion. Also it would be interesting to know whether samples coming from different slides (presumably coming from two distinct FFPE blocks) are more different than samples coming from the same slide.

The text makes several statements that are poorly justified in my opinion. E.g. the authors say that the distribution in 5g is wide and the correlation in 5h is low, and draw several conclusions based on these judgments, but I have trouble following their logic. Presumably, the distribution in 5g could be equally wide if there was no signal at all and clones were randomly distributed on the tree? In 5h, the correlation is low in comparison with what expected value? Why should 0.37 not be considered large? Can the authors provide a p-value? I believe that with the current justification, the authors are overstepping in their interpretation. Generally the impression is that the authors are using very thin data to argue very strongly for the Big Bang model and against stepwise progression, this should probably be toned down a bit, as it appears somewhat biased (particularly in the section on selection which presents very limited data).

We thank the editors and reviewers for their thorough review and for providing constructive critique on the manuscript. Please find below a detailed point-by-point response to the reviewers' concerns.

Reviewer #1

Ryser and colleagues describe an interesting multi-region sequencing study to analyze the invasion pattern of colorectal cancers. Multiple subclones are identified from mutations in a panel of targeted genes and from the spatial organization of the subclones the authors conclude that multiple subclones invade in parallel and the growth dynamics is neutral. I am, however, not sure if their data fully supports this conclusion.

Issue 1: To conclude that multiclonal invasion and neutral growth is the dominant mode of growth I believe that in a tumor the following spatial distribution of sub-clones should be observed: 1) Multiple clones should be observed both above and below the muscularis mucosae (the boundary for invasion used by the authors). 2) The genetic diversity should be similar above and below the muscularis mucosae. Only in tumor M both of the above conditions are met, in all other tumors presented in the main manuscript either there is just one subclone present both above and below the muscularis mucosae or the genetic diversity is much lower below the muscularis mucosae (e.g. in tumor H the green clone might be marginally invasive but the blue clone clearly dominates the invasive area, indicating a selective benefit.).

We agree with the reviewer in that multiclonal invasion and branching evolution should manifest itself in multiple clones that span from superficial to invasive regions, and that genetic diversity should be comparable between superficial and invasive portions of the tumors. Here, we summarize statistics across 9/12 tumors, excluding two tumors without any invasive subclones (K and C) and one tumor with only one subclone (J):

name	super	inv	span	simpson_super	simpson_inv
D	3	0	2	0.7252747	0.2091503
E	2	2	2	0.7333333	0.6985294
F	1	1	3	0.7142857	0.6916667
H	0	1	2	0.5090909	0.6013072
R	0	1	2	0.2857143	0.4642857
M	0	2	5	0.8205128	0.8583333
T	0	1	4	0.7857143	0.8161765
U	1	1	2	0.4166667	0.5604396
W	1	0	6	0.8717949	0.8750000

More precisely, we conclude the following:

- Column "span": all tumors have between 2 and 6 unique subclones (mean: 3.1) that span across between superficial and invasive regions (these data are also found in Table 1 in the main text).
- Columns "super" and "inv": there is a limited number of subclones per tumor that are exclusively superficial (mean: 0.9) or exclusive invasive (mean: 1.0) (these data are also found Table 1 in the main text).
- Columns "simpson_super" and "simpson_inv". Based on the reviewer's comments, we further computed the tumor-specific Gini-Simpson diversity index for superficial and invasive portions. This index provides a measure of subclone diversity between 0 and 1, with 1 meaning maximum diversity. We found a lack of statistically significant differences in diversity when comparing superficial and invasive regions (paired t-test, $p=0.15$).

The Gini-Simpson index was not reported in the first version of the manuscript and we thank the reviewer for suggesting this. We added the corresponding figure in the section "Fingerprints of multiclonal invasion" on page 5 (the original Fig. 5f was sent to the appendix as Fig. S6):

...tumors ... exhibited comparable levels of genotypic diversity in superficial and invasive regions (Fig 5f).

(New Fig 5f)

The reviewer further points out that there are tumors with suspected selective benefit among invading clones. We agree and refer to the paragraph “Recognizing stepwise progression” on page 7 of the manuscript where we discuss the presence of private driver mutations that suggest the presence of a selective benefit among certain subclones.

Finally, we would like to emphasize that the primary conclusion in this manuscript is about the presence of multiclonal invasion, not about neutral evolution. While our data are largely consistent with a big bang model of near-neutral or branching evolution, we completely agree with the reviewer that it does not provide a definitive delineation between the big bang and stepwise linear progression models. To clarify this distinction, we have revised the narrative, please see Issue 5 of Reviewer #2.

Issue 2: In Fig. 1 the authors do introduce two models of growth and the expected spatial patterns of subclones. It would help if they quantify the expectations of these models, and quantify their observed data, to compare and assess which tumor fits in which model. Contrary to what the authors suggest in the discussion, I believe their work could greatly benefit from explicit modelling efforts. In particular, it would be very interesting to see the predictions from spatially resolved modelling in conjunction with their spatial genetic data, similar to e.g. Sun et al., Nature Genetics 2017 and Van der Heijden et al., PNAS 2019.

We apologize if our statement may have led to confusion about the declared role of modeling and we would like to clarify this point. In fact, what we meant to convey is that the results presented in this paper do not rely on underlying parametric models that are based on assumptions about tumor growth dynamics. On the other hand, we fully agree with the reviewer that the presented data could inform additional insights through the application of mathematical models.

We previously used a modeling-based approach to provide insights into early cellular mixing patterns in colorectal tumors, see Ryser et al., PNAS, 2018. Because the model we had developed for the subclone mixing problem is not applicable to the current data set – vertical growth dynamics are central to the current manuscript – a new model first needs to be developed and calibrated. We are currently working on such a model extension and the challenge of fitting to the data at hand. It turned out to be a non-trivial and computationally expensive endeavor. Therefore, the modeling aspect exceeds the scope of the current manuscript and will be the subject of a forthcoming publication.

Based on the reviewer’s feedback, we changed the sentence in the discussion section (page 10) to:

Our analysis is non-parametric and thus not limited by modeling assumptions which can be difficult to verify experimentally^{14,15}. Nevertheless, the combination of the phylogeographic data with dynamic

mathematical models could provide valuable insights into the evolutionary dynamics of these tumors, and is subject to future research.

Issue 3: In addition it would help if the authors mention their definition of a subclone in the main manuscript.

We have added the following addendum in the main text on page 4:

...subclone, that is a group of cells sharing the same genotype.

In the methods section, we further added the following clarification on page 13:

After pre-processing, each spot was assigned a genotype, that is a binary vector indicating the presence and absence of mutations. Unique genotypes were labelled as subclones and consequently, spots that share the same genotype belong to the same subclone.

Issue 4: Are the authors planning to make their data publicly available?

Yes, the data will be published as supporting material. Actually, it should already be available to the reviewers in the csv file called "Ryser_SuppData.csv." Similarly, the code to process the data should also be available to the reviewers during the review process (a documented folder containing all code was sent for review), and will be posted to GitHub if the paper is accepted for publication.

Reviewer #2

The manuscript entitled “Minimal barriers to invasion during colorectal tumor growth” by Ryser et al. presents targeted sequencing data from small colorectal cancer samples that the authors classify as superficial or invading. By placing the tumor areas on a phylogenetic tree and observing that invading areas are polyphyletic groups, the authors argue that invasion is not the trait of a unique subclone, but that many if not all cells are capable of it.

In my opinion, the topic addressed here is of great interest to the scientific community. Much remains to be learned about the evolution of tumors, and it is important and meaningful to investigate whether the invasive phenotype may be correlated with a distinct genotype. The first step is to determine whether invading cells belong to the same clone, which the authors attempt to do here. However, while the topic is very interesting, I am not convinced by the implementation of this study. My main concern is the lack of detail in the description of the histopathological work, which is the foundation for all conclusions drawn here.

Issue 1: Pathological review and categorization of samples: the methods section contains 3 lines of text on the topic of “tumor samples”. That is not sufficient for a paper whose main conclusion is based on pathological categorization of tumor areas into invading and superficial. Vastly more detail would be needed here. How were the tumor areas classified? Presumably by a GI pathologist? Which rules did this pathologist employ? The text mentions that the muscularis mucosae was used as a dividing line, but what about tumors in which the muscularis mucosae no longer exists because the entire area between the lumen of the colon and the stroma is occupied by tumor? The dashed dividing lines in many of the histological images are concerning in my opinion. For example, in figure 2i (both slide A3 and A4), the dividing line appears to be running straight through the epithelium and sometimes it even runs within the lumen... Many clones that are categorized as “invasive” by such a casually drawn line are superficial clones in my opinion (e.g. several of the green clones in slide A3/Figure 2). Similar concerns apply to other slides, e.g. slide A3 in Figure 2B. In the absence of a very detailed description of the analysis performed by the GI pathologist, I am doubtful that many of these clones are truly invasive, and not just contiguously growing tumor on the mucosal surface. Since the conclusions of the study rest so heavily on the superficial/invading categorization, the authors need to justify their categories in much more detail.

We agree with the reviewer that the categorization of spots may be subject to interview observability and we apologize for a lack of more detailed documentation of the classification algorithms. To address the reviewer’s concerns we made amendments to the main text, and added a new paragraph to the Methods section entitled “Phenotypic classification.” Below we summarize the main points:

The classification algorithm as implemented by the pathologists is now explicitly outlined in the Methods:

- a. *In tumor sectors where the muscularis mucosae (MM) remained intact (Tis stage), cancer cells above the MM formed a Tis lesion and were labelled “superficial”.*
- b. *In tumor sectors where the MM was no longer present (T1+ stage), cancer cells were labelled “superficial” if close to the surface of the lesion (lumen), and “invasive” if in deeper portions of the tumor.*
- c. *The trajectory of the MM was indicated where still intact. In sectors with degraded MM, the most likely path of original MM was indicated; this was done for visualization purposes only and had no bearing on the spot classification.*

Regarding the implementation: in a first round, phenotypic classification of micro-dissected spots was independently performed by two board-certified pathologists (AH and DS). Disagreements were resolved through consultation with a third board-certified pathologists (ST) who specializes in GI pathology. During the revisions, based on the input from the GI pathologist, 5 spots in 3 carcinomas (H, U, W) were reclassified. Of note, this reclassification did not affect the distribution of subclones across the categories of “superficial”, “purely invasive” or “mixed”, respectively, and thus had no impact on the main results and conclusions of the manuscript.

During the revisions, and based on the input from the GI pathologist, spots that had been labeled as “stalk invasion” in the adenoma K were reclassified as pure adenoma locations. In consequence, we removed discussion of stalk invasion in the results section.

The full process description has been added in the Methods section on pages 12-13 of the manuscript.

Issue 2: Given that all clones come from a very small sampled tumor area (corresponding to one or two slides), it is not clear to me how meaningful the “early/late” classification is. Presumably, given the constrained sampling area, these could all be “late” subclones that have recently evolved in this part of the tumor. This should at least be discussed in my opinion.

We thank the reviewer for pointing this out. We agree that the early/late distinction is relative because early events may have unfolded over a much longer evolutionary time span, and the final expansion (which we are measuring here) may therefore consist of predominantly “late” subclones. As suggested by the reviewer, we added a clarification about the early/late classification as follows on page 6

We note that we only analyzed a limited number of spots in each tumor, and it is likely that the resulting phylogenies do not capture the full ancestries of the tumors. However, because the ordering of subclones on the sampled trees is the same as on a more complete tree (i.e. from the zygote to the tumor), we can nevertheless conclude that there was no evidence of later subclones being more phenotypically advanced than earlier subclones.

Issue 3: Also it would be interesting to know whether samples coming from different slides (presumably coming from two distinct FFPE blocks) are more different than samples coming from the same slide.

This is a great question, thanks for the inquiry. We looked into this by computing the pairwise distances of subclones between slides and within slides. In short, we found that with the exception of tumor C, there were no significant differences in pairwise subclone distances when comparing within vs between slide subclones (p-values from two-sided Wilcoxon test). These results have been added as Fig. S7.

New Fig S7

Issue 4: The text makes several statements that are poorly justified in my opinion. E.g. the authors say that the distribution in 5g is wide and the correlation in 5h is low, and draw several conclusions based on these judgments, but I have trouble following their logic. Presumably, the distribution in 5g could be equally wide if there was no signal at all and clones were randomly distributed on the tree? In 5h, the correlation is low in comparison with what expected value? Why should 0.37 not be considered large? Can the authors provide a p-value? I believe that with the current justification, the authors are overstepping in their interpretation.

We thank the reviewer for raising these concerns. We revisited this part of the results section to ensure that there is a clear link between figures and derived statements.

Regarding Figure 5g, we indeed meant to convey what the reviewer puts here more eloquently, i.e., that the distribution is not distinguishable from a uniform distribution. To formally test this difference between the empirical distribution and an analytic uniform distribution, we performed a two-sided Kolmogorov-Smirnov test. The null hypothesis (drawn from a uniform distribution) not rejected at standard significance levels ($p > .1$), indicating that the distribution is not statistically different from a uniform distribution.

Regarding Figure 5h, we acknowledge that different fields have different interpretations of what constitutes strong vs weak correlations. Therefore, we decided to refrain from judging the magnitude of the absolute correlation values beyond commenting on the range and median. To provide a quantitative angle to this aspect, we added the results generated in response to “Issue 3” above (new Fig. S7), that is the comparison of pairwise distances between subclones within and between slides of the same tumor. In fact, this analysis underlines that subclones within a slide are not more distinct than subclones between separate slides.

To summarize the above changes, we changed the corresponding section in the manuscript as follows:

Although contiguous subclones are expected with both stepwise progression and branching evolution, the phylogenetic relatedness of neighboring subclones can provide further delineation between the models. First, we found that the normalized genetic distance between adjacent subclones had an empirical distribution (Fig. 5h) that was not significantly distinct from a uniform distribution ($p > .1$, two-sided Kolmogorov-Smirnoff test). Thus, and in contrary to what would be expected under stepwise progression, physically adjacent subclones were essentially randomly sampled from the ancestral tree. Second, the relationship between spatial and genetic distances of spots within slides was heterogeneous, with Pearson correlations ranging from -0.2 to -0.8 (median: 0.37) across the 12 tumors (Supplementary Fig. S6). Third, we found that the pairwise distances of subclones in the same slide were similar to those of subclones from two separate slides in the same tumor. With the exception of tumor C, there were no differences between within- and between-slide pairwise distances (Supplementary Fig. S7). Taken together, these observations suggest that the co-evolution of equivalent subclones may have been preceded by early subclone mixing⁹, leading to the observed jigsaw arrangements of subclones in the final tumor as evidenced by a lack of consistent correlations between spatial and ancestral relatedness.

Issue 5: Generally the impression is that the authors are using very thin data to argue very strongly for the Big Bang model and against stepwise progression, this should probably be toned down a bit, as it appears somewhat biased (particularly in the section on selection which presents very limited data).

We thank the reviewer for this feedback. In retrospect, we agree with the reviewer that our language may have been too strong with respect to the Big Bang model. This was definitely not our intention, since the data from the current study does not provide definitive delineation between stepwise progression and the Big Bang model. We made modifications to emphasize the following main conclusions and limitations across the manuscript:

- The data directly support the presence of multiclonal invasion,
- The data are broadly *consistent* with a Big Bang model of branching evolution; however, the data do not provide conclusive evidence about whether the subclones are neutral or have a fitness advantage and therefore the data do not prove or disprove the presence of neutral evolution.
- The notion of a Big Bang growth in these samples is largely based on prior analyses of these tumors that have revealed evidence of Big Bang growth patterns. Corresponding references are provided throughout the text.

To make sure that our interpretation of the data is not misconstrued, we carefully revisited the manuscript and made a number of small word changes and refinements to clarify. Please refer to the annotated revision copy where the changes are highlighted in red.

REVIEWERS' COMMENTS:

Reviewer #1 (Remarks to the Author):

The authors have addressed most of my concerns. Only the statement regarding modelling (Issue 2) remains highly confusing. Although the authors do no computational modelling, they do introduce two distinct models in the first figure. The predictions of these models are subsequently tested experimentally. Hence their analysis is limited by modelling assumptions. To avoid confusion, I suggest the authors completely remove the statements related to Issue 2. This is only a minor point, further I can recommend publication.

Reviewer #2 (Remarks to the Author):

Thank you for the revisions - my questions have been effectively answered and I have no further comments.